# Effects of Manual Lymphatic Drainage with Mobilization and Myofascial Release on Muscle Activities during Dynamic Balance in Adults with Calf Muscle Shortening

**DOI:** 10.3390/healthcare12101038

**Published:** 2024-05-17

**Authors:** Se-Yeon Kim, Ki-Song Kim, Young-In Hwang

**Affiliations:** 1Department of Physical Therapy, Smart Healthcare Convergence Research Center, Research Institute for Basic Sciences, The Graduate School, Hoseo University, Asan 31499, Republic of Korea; 20182730@vision.hoseo.edu; 2Department of Physical Therapy, Smart Healthcare Convergence Research Center, Research Institute for Basic Sciences, College of Life and Health Sciences, Hoseo University, Asan 31499, Republic of Korea; kskim68@hoseo.edu

**Keywords:** ankle joint range of motion, fascia release, interstitial fluid drainage, manual lymphatic drainage, postural balance

## Abstract

Mobilization with movement (MWM) and myofascial release (MFR) are treatment techniques that increase ankle dorsiflexion range of motion (DFROM). Manual lymphatic drainage (MLD) facilitates waste drainage and improves soft tissue tension in peripheral tissues. To date, no studies have investigated how the combination of MLD, MWM, and MFR influences the human body. The purpose of this study is to determine how the combination of MLD, MWM, and MFR affects DFROM and balance ability. We randomly assigned 16 individuals (26 feet) to one of three groups: MWM-MFR (MR), MWM-MLD (MD), or MWM-MFR-MLD (MRD) intervention. To confirm the intervention effect of each group, DFROM was assessed using a modified lunge test, and dynamic balance was measured using a modified star excursion balance test. In the results, differences were found between the MR and MRD groups in PL and mGCM activities in the 1 section (*p* = 0.008, *p* = 0.036) and between the MD and MRD groups in mGCM activity in the 4 and 5 sections (*p* = 0.049, *p* = 0.004). We suggest that the application of MRD is the most effective intervention for increasing muscle activation of the PL and mGCM during the modified star excursion balance test.

## 1. Introduction

The ankle range of motion (ROM) is an important part of the human kinetic chain and plays an important role in postural balance. Limitations in the dorsiflexion range of motion (DFROM) can cause a significant decrease in balance ability [1]. Hoch et al. [2] reported a high correlation between anterior reach and ankle DFROM during the weight-bearing lunge test (WBLT) when the star excursion balance test (SEBT) was conducted. Proper activity of the tibialis anterior and mobility of plantar flexors must be secured for smooth DFROM. In particular, stretching the plantar flexors is important, as shortness in the plantar flexors can cause a number of pathological problems, such as limitations in DFROM, plantar fasciitis, and Achilles tendinitis [3].

Mulligan’s mobilization with movement (MWM) is based on improving mobility through continuous passive joint gliding while patients move their joints through active functional movements [4]. MWM activities significantly improve ankle DFROM and SEBT maximal reach [5,6].

Muscles exhibit functionally integrated continuity through fascia, even during different actions; thus, muscle shortening is also associated with myofascial shortening [7]. Myofascial release (MFR) is a treatment technique developed to reduce fibrous adhesion between fascia tissue layers, by increasing the length and width of myofascia by increasing and softening connective tissue [8]. Myers argued that, when MFR is performed by placing a tennis ball on the surface of the sole and applying pressure, it increases the ROM and flexibility in the toe touch test [7]. Similarly, Grieve et al. confirmed the relaxation effect of plantar fascia using tennis balls and reported a significant increase in lower limb flexibility during the sit and reach test (SRT) [9]. MFR applied to the plantar fascia also results in an increase in muscle length in the intestinal muscles, due to the continuity of the myofascia, which leads to an increase in DFROM [7].

Manual lymphatic drainage (MLD) is a treatment technique that reduces inflammation, edema, and pain by relaxing sympathetic nerves without irritation through mild skin elongation and by promoting the drainage of body fluids [10]. Physical pressure and rhythmic and mechanical stretching of soft tissue through MLD improves peripheral tissue tension and facilitates waste drainage [11]. MLD may also offer therapeutic benefits in passive and active ROM [12].

MWM is a method mainly used for increasing DFROM. In a literature review study examining the effect of MWM, the majority of studies reported that DFROM was significantly improved when MWM was applied [13]. In addition, a previous study reported that not only DFROM but also dynamic balance ability increased when MWM and MFR were applied together [14]. DFROM has a great effect on dynamic balance [15]. In this study, MWM and MFR are positive for improving DFROM and dynamic balance ability, but it is thought that there will be changes in body tissue fluid due to the movement of joints and fascia in the body, and these changes will cause limitations in improving ROM and balance ability.

Therefore, in this study, we compare the group with MWM and MFR, the group with MWM and MLD, and the group with MWM, MFR, and MLD to identify which intervention has the most positive effect on DFROM and dynamic balance ability.

## 2. Materials and Methods

### 2.1. Design and Participants

This study was conducted at Hoseo University through poster recruitment for adults in their twenties living in Cheonan and Asan, Korea, from October to December 2022, after obtaining permission from the Hoseo University Bioethics Committee (1041231–221013-HR-153). Post-test was conducted immediately after the intervention to evaluate the immediate effect of the intervention. All experimental procedures were performed in accordance with the Declaration of Helsinki. All subjects provided informed consent.

Those with heart or nervous system-related diseases and those with a history of previous surgery on musculoskeletal structures (bones, joint structures, nerves) of both lower extremities were excluded. Inclusion criteria included patients who had a passive ankle dorsiflexion with a range of less than 8° while the knee was extended in the prone position; who, after adjusting for the neutral position under the subtalar joint, showed an ankle dorsiflexion with the knee flexion greater than that with knee extension by 5° or more; and in whom the difference in the length of both legs was less than 2 cm [15].

The left and right lower limbs of the subjects were viewed as independent feet. A total of 16 people (26 feet) were recruited. One subject returned home for personal reasons. Thus, the total number of participants, excluding those who gave up halfway, was 16 people (26 feet). Participants were assigned to one of three groups according to the intervention to determine which intervention is the most effective for improving DFROM in adults with shortened calf muscles: MWM-MFR (MR), MWM-MLD (MD), or MWM-MFR-MLD (MRD). Participants were assigned to the MR group (*n* = 5, feet = 8), MD group (*n* = 6, feet = 9), or MRD group (*n* = 5, feet = 9). The general characteristics of the subjects are listed in Table 1. Participants were assigned to one of three groups listed above.

A randomized controlled trial with a factorial design was employed. Randomization was performed using Excel’s (2016 version) ‘=RANDBETWEEN’ function. A single-blind design was employed where each subject was blinded to the differences in intervention methods between groups.

An assistant prevented the researcher from intervening in the subject’s group assignment and test measurements.

The sample size was calculated using G*Power 3.1 software [16]. A sample of 34 people was required based on an effect size of 0.565 resulting from the pilot study with a significance level of 0.05 and power of 80% in the analysis of covariance (ANCOVA). Accordingly, a sample of 34 people was deemed necessary [16], and this study assessed 34 feet for eligibility but only 26 feet were included and 8 feet were excluded. The flowchart of the study was followed in Figure 1. 

### 2.2. Test Procedures and Instrumentation

#### 2.2.1. Ankle Dorsiflexion Range of Motion

A modified lunge test was used to determine each subject’s active ankle DFROM. DFROM measurement via the WBLT is an accurate way to identify the maximum ROM during functional activities such as walking and running [17]. In a review that compared the reliability of DFROM measurements, WBLT was a clinically useful evaluation method, showing high reliability within and between measurements [18].

The modified lunge test is a simpler measurement than the conventional WBLT, which measures the DFROM of the ankle located at the back during the lunge (Figure 2) [19]. Subjects, after standing upright in a comfortable position, move into a lunge position by extending the opposite foot as far as possible. Participants underwent two trials of the modified lunge test to familiarize themselves before DFROM measurements were taken.

The iHandy Level app, an app from iPhone 11 inclinometer applications (iPhone, Apple, Cupertino, CA, USA), shows high reliability and validity within and between evaluators when measuring ROM [20]. Thus, WBLT was used to evaluate DFROM in the current study. The test was repeated three times, and the average value was used in statistical analyses.

#### 2.2.2. Dynamic Balance

The modified SEBT (mSEBT), a simplified version, was performed to evaluate the dynamic balance of the study participants. Based on a previous study [21], the maximal reaching distance in three directions was used to determine the subject’s dynamic balance ability: anterior (Ant), posteromedial (PostM), and posterolateral (PostL). The mSEBT has been proven to be an excellent evaluation method for dynamic balance in terms of its reliability in healthy people [22].

The mSEBT was conducted with reference to the work by Gribble et al. [23], who calculated the average value for kinematic movement by referring to 1 section to Ant forward, 2 section to Ant backward, 3 section to PostM forward, 4 section to PostM backward, 5 section to PostL forward, and 6 section to PostL backward (Figure 3). Robinson et al. performed the mSEBT six times and reported that the stability of the limbs increased from the fourth measurement on, with significantly reduced measurement error and enhanced reliability [24]. Thus, we first administered four practice tests. Then, the actual test was conducted three times. The maximal reaching distance was measured after the test and then was divided by the structural leg length of the subject and multiplied by 100 to obtain a normalized value for statistical analysis [23].

Inertial measurement unit (IMU) 3D myomotion (MyoMotion, Noraxon, Scottsdale, AZ, USA) was used to analyze the kinematic variables of the lower extremities during dynamic balance measurement (Figure 4). This unit employs a sensor that can analyze body segmentation movements in three dimensions when movements appear and has an accuracy of 1.2° for dynamic balance evaluations [25]. Seven IMU sensors were attached to the center of the left and right posterior superior iliac spine, and one to the left and right thigh, shank, and foot (Figure 5) [25]. During the measurement, the sampling frequency of the IMU sensor was set to 200 Hz, and the instantaneous change in the angle of the lower extremity joint was recorded [25].

Surface electromyography (EMG) (Ulium EMG, Noraxon) was used to investigate changes in ankle muscle activity during dynamic balance measurements (Figure 4). The investigated muscles were the tibialis anterior, peroneus longus (PL), the lateral gastrocnemius (lGCM), and the medial gastrocnemius (mGCM). EMG attachment locations for each muscle were based on the recommendations of the Surface Electromyography for the Non-Invisible Assessment of Muscle project (Figure 5). The raw data signal from the EMG was filtered through a band between 10 and 500 Hz [26]. Electrodes were attached in the vertical direction of muscle resolution; the distance between electrodes was maintained at 2 cm to minimize crosstalk [27].

Before the test, the maximum voluntary isometric contraction (MVIC) of each muscle was measured, and the lower extremity muscle activity data obtained during dynamic balance was normalized and compared. The MVIC measurement of each muscle was based on the frequency muscle strength test posture described by Kendall et al. [28]. MVIC data were collected for 5 s, of which 3 s were used for normalization (excluding the first second and the last second); a 5 s measurement and a 5 s break were repeated three times, and the maximum value was used for analysis [29].

The FDM-S system platform (Zebris Medical GmbH, Isny im Allgäu, Germany) was employed to analyze changes in the center of pressure during dynamic balance measurements (Figure 4). This is an instrument that can measure static and dynamic movements of the load distribution under the feet while standing or walking [30].

### 2.3. Test Procedures

The MR group conducted MWM for 5 min and MFR for 5 min, for a total of 10 min. The MD group conducted MWM for 5 min and MLD for 15 min. The MRD group conducted MWM for 5 min, MFR for 5 min, and MLD for 10 min, for a total of 20 min. It was applied once to investigate immediate effects. Each intervention was applied as follows.

In MWM, the subjects placed their foot on a chair (height: 40 cm) before the intervention and performed maximal dorsiflexion in a closed-chain state. After that, they used the inelastic tape to allow for posteromedial gliding to appear while walking (Figure 6a). The intervention was to walk for 5 min on a treadmill with a speed of 4.5 km/h and a slope of 6° (Figure 6b) [16].

In MFR, it was relaxed by placing a tennis ball on the plantar fascia and calf muscle fascia and then applying pressure with weight (Figure 6c,d). The pressure was applied for 2 min to the plantar fascia and 3 min to the calf muscle fascia, for a total of 5 min.

MLD was performed on the lymph node of the lower extremity (Figure 6e). MLD was applied by a researcher. The application of MLD followed Dr. Vodder’s methods, starting by gentle effleurage of the leg, followed by applying it sequentially to the thighs, inguinal lymph nodes, knees, lower leg, Achilles tendon, ankle, dorsum of the foot, lymph sea, and transverse arch, finishing with effleurage. MLD was applied by a researcher who has completed 40 h practice with the Vodder methods and, to minimize variations in technique, only one researcher applied MLD throughout this study.

### 2.4. Statistical Analyses

Statistical analysis was conducted using SPSS statistics (ver. 20.0; IBM Corporation, Armonk, NY, USA). Since the normality was not confirmed in the Shapiro–Wilk test, the ACOVA could not be used. Thus, the Kruskal–Wallis test, a non-parametric test, was conducted, with a significance level (α) set at 0.05. Furthermore, post hoc tests were performed using Bonferroni correction, with a significance level (α) set at 0.05. Additionally, one-way analysis of variance (ANOVA) was conducted to determine whether there were significant differences among general characteristics between groups. A paired t-test was used to confirm the comparison results before and after the intervention of the maximal reaching distance for each group. For all statistical analyses, the significance level (α) was set to 0.05.

## 3. Results

In the Kruskal–Wallis tests, to determine the effect of intergroup intervention on the dependable variable, significant differences were found in DFROM (*p* = 0.048) and the 3 section of pelvic rotation (*p* = 0.024) among kinematic data. However, there were no significant differences in the post hoc test for either of them. In the 1 section, there was a significant difference of PL muscle activity (*p* = 0.011). The 1, 4, and 5 sections of mGCM of muscle activities were significantly different (*p* = 0.020, *p* = 0.021, *p* = 0.003, respectively).

In the post hoc test, a significant difference was found between the MR and MRD groups (*p* = 0.008). In the 1 section, there was a significant difference in mGCM activity between the MR and MRD groups (*p* = 0.036); and in the 4 section, there was a significant difference in mGCM activity between the MD and MRD groups (*p* = 0.049). In the 5 section, a significant difference was found between the MR and MD groups (*p* = 0.033), and between the MR and MRD groups (*p* = 0.004) (Table 2).

A paired t-test was conducted to determine the effect of intervention on the maximal reaching distance during mSEBT. There were no significant differences in the MR group. Significant increases were found in the MD group when reaching the PostL (*p* = 0.045) direction, and in the MRD group in all directions of Ant (*p* = 0.019), PostM (*p* = 0.015), and PostL (*p* = 0.027) (Table 3).

## 4. Discussion

This study aimed to identify which intervention application had the most positive effect on the study results in terms of DFROM and dynamic balance ability when comparing the MR, MD, and MRD groups. As a result of the study, the MRD group showed the greatest improvement with the application of MWM and MLD with MFR.

MRD groups showed a significant increase in DFROM in between-group comparisons, suggesting that the simultaneous application of MFR and MLD increases the muscle length of the shortened calf muscles. Self-myofascial release (SMR) is a treatment technique that relaxes the calf by increasing muscle tension through pressure applied to the myofascia, based on the function of Golgi tendon organ (GTO) [31]. When applying SMR, passive stretching pressure is transmitted to the muscles, the activity of the muscle spindle is suppressed, and the GTO is stimulated, thus resulting in muscle relaxation. In previous studies, SMR has been introduced as a treatment technique for restoring shortened muscle length and improving function through the relaxation of these muscles [31].

Chang et al. reported a significant negative correlation between ankle DFROM and stiffness of the gastrocnemius–Achilles tendon [32]. When SMR was performed on the calf, ankle DFROM increased through WBLT after intervention, and the stiffness of the gastrocnemius–Achilles tendon significantly decreased [32]. In another study, the flexibility of the biceps femoris and the DFROM increased when MFR was performed on plantar fascia [9]. Based on the results of these studies, it is believed that the MFR in this study also increased DFROM by relaxing the calf muscle via the self-suppression mechanism of the GTO.

In addition to the effect of MFR, it may be suggested that the application of MLD promotes interstitial drainage to create a synergy effect. Cao et al. investigated differences in fibroblast proliferation, hypertrophy, and cytokine secretion over time and the intensity of MFR application [33]; they showed that the dry weight and cytokine secretion increased when the MFR strength increased from 3% to 12%, and cytokine secretion significantly increased as the MFR application time increased to 5 min [33]. Changes in dry weight observed without changes in DNA content or total intracellular protein suggest that the production of extracellular matrix proteins may be elevated. This is due to the fact that the application of MLD effectively drains moisture, proteins, and other substances that accumulate in the interstitium. However, in the post hoc test, there was not a significant difference between the MD and MRD groups. We thought that there was no significant difference in DFROM between the MD and MRD groups due to the relatively small sample size.

Among the kinematic data of this study, the MD group showed the largest increase in the 3 section of pelvic rotation compared to MR and MRD groups (*p* = 0.039). This increase was believed to result from compensating for pelvic over-rotation instead of flexion of the hip or knee joints to achieve longer reach. However, in the post hoc test there was no significant difference between the MR and MD groups (*p* = 0.054). Notably, Gribble et al. reported that the kinematic dynamic balance measured via SEBT is influenced by sagittal plane movement of the hip and knee joints, i.e., flexion [34]. A comparative analysis of kinematic data between healthy subjects and patients with chronic ankle instability (CAI) confirmed that healthy subjects reached further through the flexion of the hip and knee joints [34]. Given that the angle of pelvic rotation was significantly greater in the MD group than in the other groups, it is assumed that pelvic rotation was used more as opposed to flexion of the hip and knee joints during mSEBT.

A comparative analysis of kinematic data of CAI subjects with controls when SEBT was performed in the direction of anteromedial, medial, and PostM showed significant pelvic rotation in the CAI group [35]. In the current study, no measurement was made on the anteromedial during mSEBT; additionally, PostM was analyzed by dividing it into forward and backward directions. Notably, a previous study analyzed PostM in one direction; thus, we do not believe that we can compare our results to those findings [35]. As no studies have analyzed the kinematic data of pelvic rotation by dividing PostM into forward and backward at the time of mSEBT, it is believed that, if the kinematic variables are analyzed by dividing mSEBT into six sections in future studies, a detailed plan for the dynamic balance of people with poor balance ability could be better implemented.

Our results demonstrate the effectiveness of MRD. The most significant improvements were observed in the 1 section of PL, and the 1, 4, and 5 sections of mGCM, indicating significant enhancement in muscle activity following MFR due to the physiological mechanisms of MLD. As mentioned earlier, MFR relaxes muscles by passively applying pressure to the myofascia [36]. Previous studies that investigated EMG changes after MFR reported that muscle activity during exercise [37] and rest [38] significantly decreased; our results are similar to those in that muscle activity decreased significantly in the MR group in the 1 section.

In addition, the MRD group showed different results from the MD group in the 4 and 5 section of mGCM; specifically, the change in the internal environment of the myofascia by the additional application of MLD effectively drained the interstitial substances produced or enhanced after MFR. The results of previous studies have shown that the distribution of pressure regulating the interstitial flow in soft tissues is most affected by large movements of the musculoskeletal system [39]. Increased interstitial flow stimulates fibroblasts, the basic cells of the myofascia, to promote differentiation into myofibroblasts and increase the production of collagen and other factors related to fibrosis [40]. This consequently increases the tension present between the cell and the extracellular matrix, which increases the size and flow of the cells [41]. Due to the fact that the MWM and MFR conducted in this study corresponded to large movements of the musculoskeletal system, it is believed that the amount of collagen and other substances present in the interstitium, as well as the size of the cells, increased. The application of MLD after MWM and MFR is considered to effectively discharge substances accumulated in the interstitium during MWM and MFR. Furthermore, the MD group saw a significant decrease in the 5 section of mGCM compared to the MR group, even though there was a significant increase of maximal reach distance in the PostL direction within the group. This is considered to demonstrate the efficiency of reaching further with less muscle effort when applying MLD. However, no study has investigated the mechanism by which MLD conducted after MFR increases muscle activation; thus, further research is necessary.

All of the maximal reaching distances in the Ant, PostM, and PostL directions of the MRD group significantly increased. The value of the maximal reaching distance of SEBT is an indicator of dynamic balance; thus, the further the distance reached, the better the dynamic balance ability [24]. To ensure that a change has occurred in the actual maximal reaching distance of SEBT, the value normalized to the leg length of the subject must increase by at least 6–8% [42]. The increase of more than 8% in the PostM direction found in our study means that the intervention of MLD after MWM and MFR effectively enhances the dynamic balance in the PostM direction during mSEBT. Hertel et al. reported that the direction of PostM arrival is the direction that best represents the overall performance of SEBT in limbs with or without CAI [21]. Therefore, the MRD group results in our study showed improvement in the overall dynamic balance ability in several directions.

This change in the MRD group is rather encouraging, but research into the effects of MFR and MLD application on kinematic and kinetic problems is still insufficient, which makes it difficult to explain the change in the MRD group. Here, we provide several potential explanations, based on the physiological mechanisms of the human body. However, our results suggest that further investigation of the MRD approach, allowing for change in the interstitium substances and the movement of tissue fluid between the interstitium and lymphatic vessels, will provide insight into how MFR and MLD react and affect kinematics/kinetics.

The limitations of this study are as follows. First, sample size of the experiment is small and it is difficult to generalize the research results to women, as the distribution of sex was biased toward men. However, because the experiment was conducted by selecting people with shortened calf muscles, expectations similar to this study can be expected for those with the same conditions. Second, our results may have been affected by temporal weight, as the intervention time applied between the groups differed. In future studies, it will be necessary to set the same intervention time to mediate this limitation. Third, this study did not include a blinded test. Fourth, only younger age groups were included in this study, so it was not possible to evaluate older age groups. Fifth, viewing the subjects’ feet as independent entities may have limited the study. Additionally, this study was a pilot study in which the calculated minimum sample size was not reached. In the future, other studies will need to supplement these points for future research.

## 5. Conclusions

This study assessed changes in DFROM in the modified lunge position, as well as kinematic and kinetic data during mSEBT when MR, MD, and MRD groups were randomly assigned to healthy adults with short calf muscles, and interventions were applied.

The MRD group demonstrated significant and positive changes in muscle activity and maximal reaching distance during mSEBT. Consequently, the application of MWM, MFR, and MLD could be effective for improving PL and mGCM activity to enhance dynamic balance during mSEBT among young healthy adults with shortened calf muscles. Since the sample size was small in this study, it is thought that more reliable results will be obtained if the same experiment is conducted with an increased sample size in the future.

## Figures and Tables

**Figure 1 healthcare-12-01038-f001:**
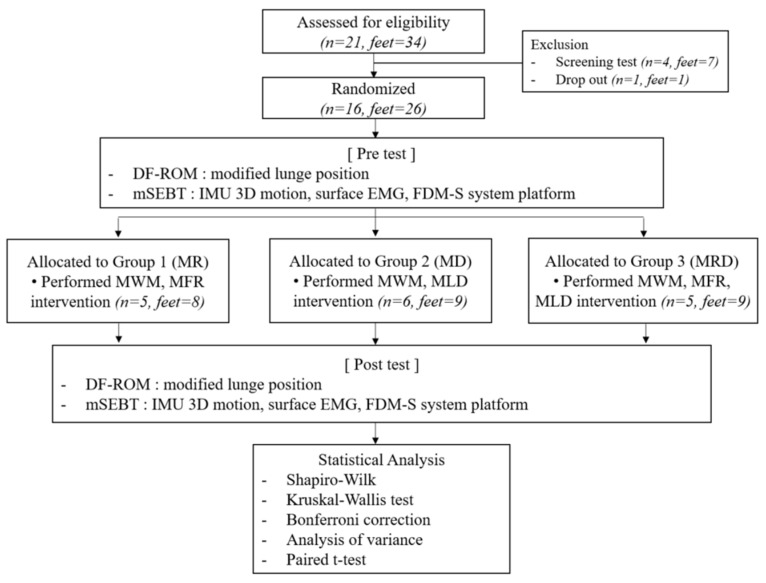
Flowchart of the study.

**Figure 2 healthcare-12-01038-f002:**
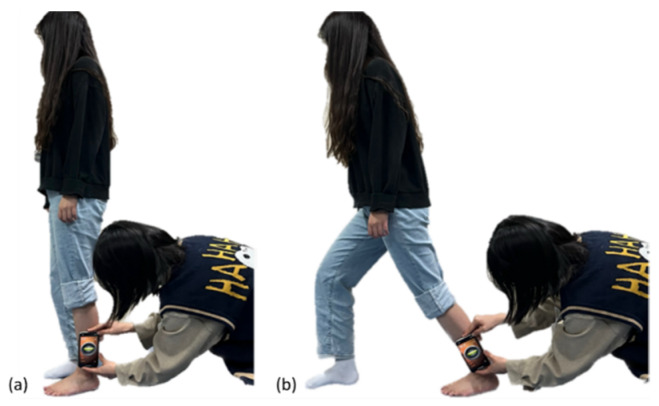
Modified lunge test. (**a**) Pre-test; (**b**) post-test for ankle dorsiflexion range of motion.

**Figure 3 healthcare-12-01038-f003:**
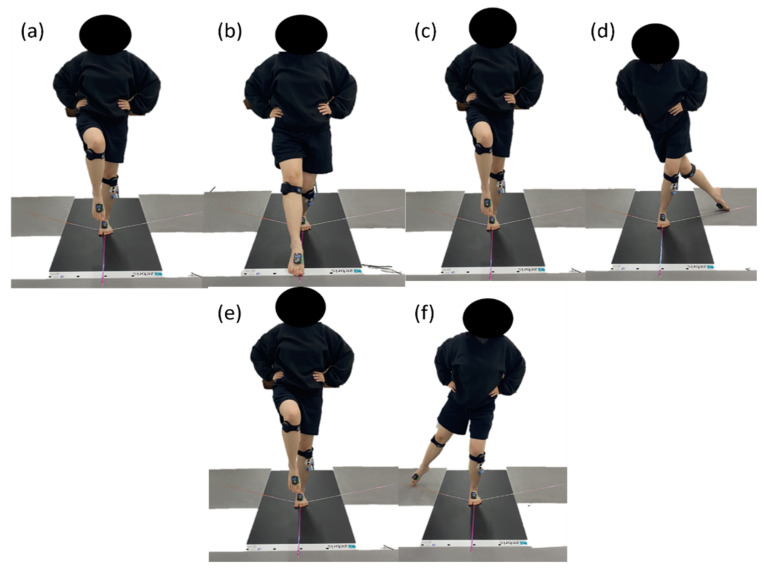
Modified star excursion balance test: All participants start at (**a**) staring position as a single leg stance; (**a**,**b**) 1 section, Ant forward; (**b**,**c**) 2 section, Ant backward; (**c**,**d**) 3 section, PostM forward; (**d**,**e**) 4 section, PostM backward; (**e**,**f**) 5 section, PostL forward; (**f**,**a**) 6 section, PostL backward.

**Figure 4 healthcare-12-01038-f004:**
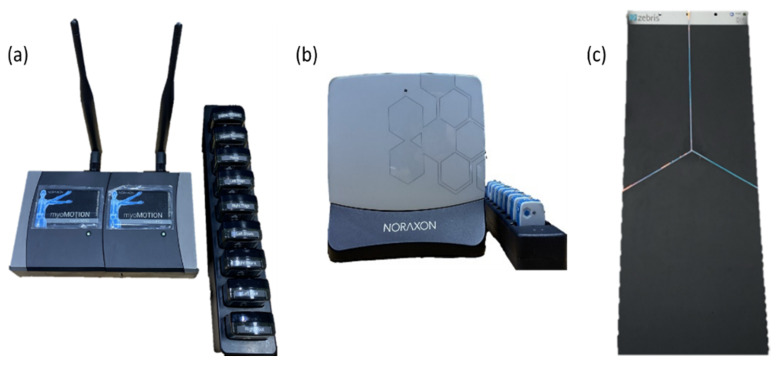
Measuring equipment. (**a**) Inertial measurement unit 3D myomotion. (**b**) Electromyography. (**c**) FDM-S system platform.

**Figure 5 healthcare-12-01038-f005:**
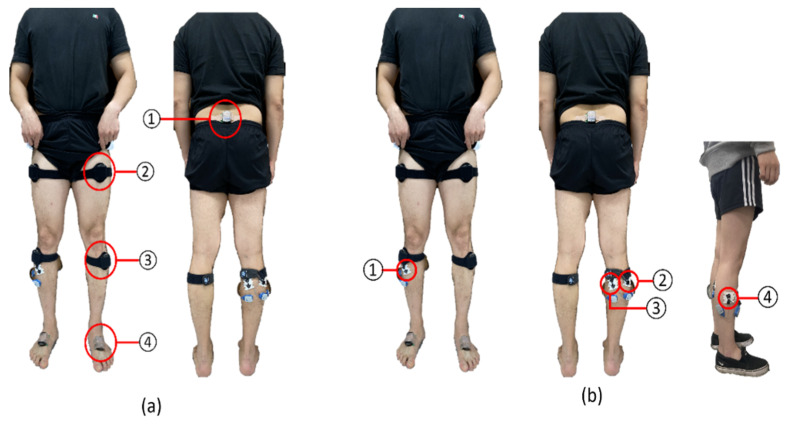
Sensor attachment site. (**a**) attachment of inertial measurement unit 3D myomotion sensor. (**a**) ① Between both posterior superior iliac spines. ② Both thighs. ③ Both shanks. ④ Both feet. (**b**) Attachment of electromyography sensor. ① Tibialis anterior. ② Lateral gastrocnemius. ③ Medial gastrocnemius. ④ Peroneus longus.

**Figure 6 healthcare-12-01038-f006:**
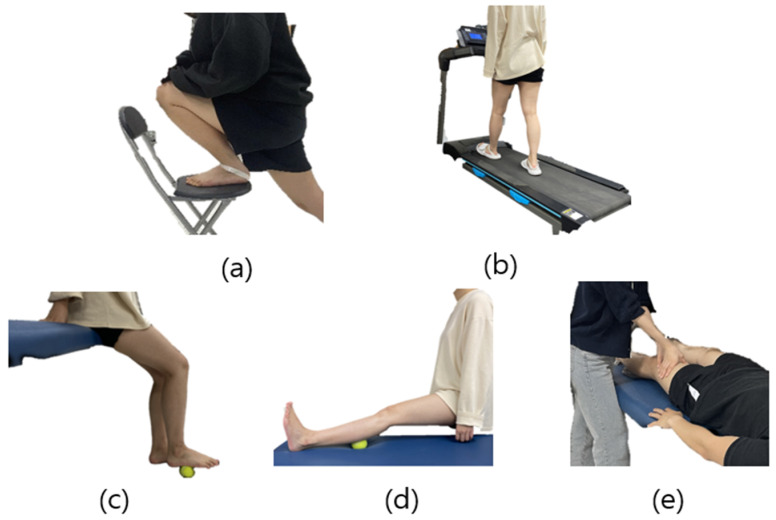
Intervention process: (**a**) posteroinferior gliding using non-elastic taping, (**b**) inclined treadmill gait using non-elastic taping, (**c**) intervention of plantar fascia myofascial release with tennis ball, (**d**) intervention of triceps surae muscle myofascial release with tennis ball, (**e**) intervention of lower extremity manual lymphatic drainage.

**Table 1 healthcare-12-01038-t001:** General characteristics of subjects in MR, MD, and MRD group (*n* = 16, feet = 26).

	MR Group (Mean ± SD)	MD Group (Mean ± SD)	MRD Group (Mean ± SD)	*p*
Age (years)	22.63 ± 1.51	24.44 ± 3.17	22.56 ± 0.53	0.095
Height (m)	1.67 ± 0.06	1.73 ± 0.07	1.73 ± 0.10	0.192
Weight (kg)	72.48 ± 9.22	71.46 ± 10.80	68.94 ± 14.60	0.817
BMI (kg/m^2^)	25.86 ± 1.93	23.80 ± 2.66	22.81 ± 3.40	0.090
Leg length (cm)	84.71 ± 4.04	88.53 ± 3.85	87.59 ± 6.10	0.259
Sex (%)				
Male	7 (87.5%)	7 (77.8%)	7 (77.8%)	
Female	1 (12.5%)	2 (22.2%)	2 (22.2%)	
Shortening side				
Right	4 (Both = 3)	4 (Both = 3)	4 (Both = 4)	
Left	4 (Both = 3)	5 (Both = 3)	5 (Both = 4)	
Dominant foot				
Right	8 (100.0%)	7 (77.8%)	9 (100.0%)	
Left	0 (0.0%)	2 (22.2%)	0 (0.0%)	

**Table 2 healthcare-12-01038-t002:** Comparison of changes after interventions in the MR, MD, and MRD groups (*n* = 16, feet = 26).

				Mean ± SD					
Variable	Section		MR (*n* = 8)	MD (*n* = 9)	MRD (*n* = 9)	χ2	*p*	Between Groups	Adjusted *p*
DFROM (°)		Pre	27.62 (5.30)	29.67 (2.67)	29.57 (3.02)	6.06	0.048 *	MR vs. MD	0.200
								MR vs. MRD	1.000
		post	28.45 (5.27)	29.66 (2.13)	30.67 (3.20)			MD vs. MRD	0.060
Pelvic rotation (°)	3	Pre	5.25 (2.37)	8.88 (2.38)	7.36 (2.90)	6.49	0.039 *	MR vs. MD	0.054
								MR vs. MRD	1.000
		Post	6.70 (2.75)	13.25 (3.67)	8.95 (3.06)			MD vs. MRD	0.146
PL (%)	1	Pre	54.71 (35.9)	66.87 (12.78)	60.92 (26.04)	9.01	0.011 **	MR vs. MD	0.352
								MR vs. MRD	0.008 **
		post	46.82 (21.74)	76.88 (20.31)	85.17 (41.83)			MD vs. MRD	0.417
mGCM (%)	1	Pre	38.60 (17.06)	57.75 (17.22)	47.82 (14.82)	7.80	0.020 *	MR vs. MD	1.000
								MR vs. MRD	0.036 *
		Post	28.72 (8.54)	47.14 (12.55)	53.62 (12.97)			MD vs. MRD	0.068
	4	Pre	68.53 (41.39)	94.44 (71.61)	60.13 (22.62)	7.71	0.021 *	MR vs. MD	1.000
								MR vs. MRD	0.052
		Post	47.12 (10.50)	66.65 (18.62)	82.21 (43.39)			MD vs. MRD	0.049 *
	5	Pre	32.40 (17.21)	45.39 (9.23)	39.49 (15.35)	11.70	0.003 **	MR vs. MD	0.033 *
								MR vs. MRD	1.000
		Post	35.88 (15.56)	30.33 (6.81)	53.95 (32.14)			MD vs. MRD	0.004 **

Values are presented as mean ± standard deviation; MR: mobilization with movement–myofascial release group; MD: mobilization with movement–manual lymphatic drainage group; MRD: mobilization with movement–myofascial release–manual lymphatic drainage group; DFROM: ankle dorsiflexion range of motion; PL: peroneus longus; mGCM: medial gastrocnemius; χ2: chi-square; Significant difference: * *p* < 0.05, ** *p* < 0.017.

**Table 3 healthcare-12-01038-t003:** Comparison of the maximal reaching distances within MR, MD, and MRD groups (*n* = 16, feet = 26).

		Mean ± SD (%)			
		Pre	Post	t	*p*
MR	Ant	89.69 ± 7.27	86.70 ± 5.97	1.53	0.171
	PostM	80.91 ± 9.50	84.22 ± 7.97	−1.95	0.093
	PostL	96.21 ± 11.96	97.62 ± 14.75	−0.31	0.767
MD	Ant	84.71 ± 9.28	90.55 ± 14.93	−1.82	0.106
	PostM	82.77 ± 13.74	87.23 ± 11.15	−1.25	0.246
	PostL	87.68 ± 11.88	97.15 ± 13.70	−2.37	0.045 *
MRD	Ant	91.63 ± 12.14	93.68 ± 10.92	−2.94	0.019 *
	PostM	83.06 ± 9.87	89.88 ± 11.70	−3.08	0.015 *
	PostL	100.28 ± 13.52	104.37 ± 16.27	−2.71	0.027 *

* *p* < 0.05

## Data Availability

Dataset available upon request from the authors.

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
