# Peer review of "Effects of Manual Lymphatic Drainage with Mobilization and Myofascial Release on Muscle Activities during Dynamic Balance in Adults with Calf Muscle Shortening"

_healthcare, 2024, doi:10.3390/healthcare12101038_

Round 1

Reviewer 1 Report

Comments and Suggestions for Authors

The authors have carried out an interesting randomized controlled trial with a fractional factorial design. Although there are data that must be implemented to facilitate understanding of the manuscript, the authors have made a broad and detailed introduction to the topic. Although I would like to present some methodological issues

Methods

-The authors should make the inclusion criteria of the participants clearer. I haven't understood them.

-After calculating the sample size, the selected sample is not understood. The selected sample does not match the calculated one. It also does not seem to match the number of feet evaluated. The authors present a small sample.

-The authors must describe how the randomization of individuals to the study groups occurred.

-The authors should describe the interventions carried out for the different groups.

-Authors should indicate whether the rater was blinded and how this was done.

-The authors should indicate the reliability of the tools used to measure the different variables.

Statistical Analyses

-The authors could calculate minimal detectable change (MCD)

- Losses in the sample (there is one) an intention-to-treat analysis should be performed to include them in the results

Author Response

Thank you for the reviewer. 

It helped a lot to make better this paper. 

Authors has attached the file. 

Reviewer 1)

The authors have carried out an interesting randomized controlled trial with a fractional factorial design. Although there are data that must be implemented to facilitate understanding of the manuscript, the authors have made a broad and detailed introduction to the topic. Although I would like to present some methodological issues

Methods

1) The authors should make the inclusion criteria of the participants clearer. I haven't understood them.

 Changed in accordance with your opinion.

Inclusion criteria included patients who had a passive ankle dorsiflexion with range of less than 8° while the knee was extended in the prone position; who, after adjusting for the neutral position under the subtalar joint, showed an ankle dorsiflexion with the knee flexion greater than that with knee extension by 5° or more; and in whom the difference in the length of both legs was less than 2cm.

2)-After calculating the sample size, the selected sample is not understood. The selected sample does not match the calculated one. It also does not seem to match the number of feet evaluated. The authors present a small sample.

-> The sample size was calculated using G*Power 3.1 software. A sample of 34 people was required based on an effect size of 0.565 resulting from the pilot study with a significance level of 0.05 and power of 80% in the Analysis of covariance (ANCOVA). Accordingly, a sample of 34 people was deemed necessary, and this study was assessed for 34 feet for eligibility but only 26 feet were included, and 8 feet were excluded-> Since the sample size was obtained based on the G*Power software, there is no study on which the sample size calculation was based.

3) The authors must describe how the randomization of individuals to the study groups occurred.

-> Randomization was performed using Excel's '=RANDBETWEEN' function.

4) The authors should describe the interventions carried out for the different groups.

Thank you for the good point.

The intervention for each group was applied as follows.

- the MR group conducted MWM for 5 min and MFR for 5 min, for a total of 10 min. The MD group conducted MWM for 5 min and MLD for 15 min. The MRD group conducted MWM for 5 min, MFR for 5 min, and MLD for 10 min, for a total of 20 min. It was applied once to investigate immediate effects. Each intervention was applied as follows.

5) Authors should indicate whether the rater was blinded and how this was done.

 Participants were randomly assigned to each group and were single-blinded to the differences in intervention methods between groups.

Randomization was performed using Excel's '=RANDBETWEEN' function.

6) The authors should indicate the reliability of the tools used to measure the different variables.

DFROM : The 124-6 Line, mSEBT : The 142-4 Line

Reviewer 2 Report

Comments and Suggestions for Authors

This article aims to compare which is better for reducing calf muscle shortening and dynamic balance: mobilisation with movement (MWM) and myofascial relaxation (MFR), or MWM and manual lymphatic drainage (MLD), or MWM, MFR and MLD.

Initially, the article seems interesting, but I would like to make some comments to the authors:

- Perhaps a title that included the other variable studied (dorsiflexion range of motion) and not "kinematic, kinetic data and muscle activities" would be more appropriate.

- Lines 32-34: Normally the reference number accompanies the authors. It would be more appropriate to use "Hoch et al. [2] reported...". This is also repeated several times in the Discussion section.

- The text states that 17 people were recruited (lines 92-93), while Figure 1 states 21. There should be consistency in the information on both sites.

- I think the sentence in line 101 is superfluous.

- Which study was used as the basis for the sample size calculation?

- Sample size calculation is not clear. First say that 66 people are needed and then 34. Furthermore, they say that this is the number of feet recruited in the study (but 8 feet were already excluded before the sample was randomised). This section needs to be clarified.

- Figure 1: the image lacks sharpness. Also, the post-test should state how many were assessed in each group after the intervention, not the statistical tests used to analyse the data.

- Table 1: I think the authors should indicate the p-value to know whether or not there were differences between groups. Then we would know whether they were homogeneous or not.

- Line 174: what does "500 Hz24" mean? 24 is a reference?

- To assess the MVIC they do 3 repetitions, but they do not indicate the rest time between them.

- References to Figure 6 are superfluous (lines 207, 210 and 213).

- There is a lack of a reference to guide the treatment of MLD applied (lines 213-214).

- Table 2 seems to have been made only with the data that was significant, is this the case? Why?

- The References section must be adapted to the journal's standards: journal name in abbreviated italics and year of publication in bold.

- Only 9/42 are references from the last 5 years. Could it be more up-to-date?

Comments on the Quality of English Language

I consider the English Language used to be correct, but I am not an expert in this language to judge it properly.

Author Response

Thank you for the reviewer. 

It helped a lot to make this paper. 

Authors attached the file to response. 

This article aims to compare which is better for reducing calf muscle shortening and dynamic balance: mobilisation with movement (MWM) and myofascial relaxation (MFR), or MWM and manual lymphatic drainage (MLD), or MWM, MFR and MLD.

Initially, the article seems interesting, but I would like to make some comments to the authors:

  1. Perhaps a title that included the other variable studied (dorsiflexion range of motion) and not "kinematic, kinetic data and muscle activities" would be more appropriate

Effects of manual lymphatic drainage with mobilization and myofascial release on dorsiflexion range of motion, kinematic data and muscle activities during dynamic balance in adults with calf muscle shortening

  1. Lines 32-34: Normally the reference number accompanies the authors. It would be more appropriate to use "Hoch et al. [2] reported...". This is also repeated several times in the Discussion section.

-> Thank you for your good point. Changed in accordance with your opinion.

  1. The text states that 17 people were recruited (lines 92-93), while Figure 1 states 21. There should be consistency in the information on both sites.

-> Thank you for the good point. Changed in accordance with your opinion .

  1. I think the sentence in line 101 is superfluous.

-> Thank you for your good point. Deleted according to your opinion.

  1. Which study was used as the basis for the sample size calculation?

-> A sample of 34 people was required based on an effect size of 0.565 resulting from the pilot study with a significance level of 0.05 and power of 80% in the Analysis of covariance

  1. Sample size calculation is not clear. First say that 66 people are needed and then 34. Furthermore, they say that this is the number of feet recruited in the study (but 8 feet were already excluded before the sample was randomized). This section needs to be clarified.

-> The authors required the effect size from the results of the pilot test and calculated the sample size .

The sample size was calculated using G*Power 3.1 software. A sample of 34 people was required based on an effect size of 0.565 resulting from the pilot study with a significance level of 0.05 and power of 80% in the Analysis of covariance (ANCOVA). Accordingly, a sample of 34 people was deemed necessary, and this study was assessed for 34 feet for eligibility but only 26 feet were included, and 8 feet were excluded-> Since the sample size was obtained based on the G*Power software, there is no study on which the sample size calculation was based.

  1. Figure 1: the image lacks sharpness. Also, the post-test should state how many were assessed in each group after the intervention, not the statistical tests used to analyze the data.

-> Authors added assessments of the post test

  1. Table 1: I think the authors should indicate the p-value to know whether or not there were differences between groups. Then we would know whether they were homogeneous or not.

-> Thank you for your good point.

Authors added p-value.

  1. Line 174: what does "500 Hz24" mean? 24 is a reference?

->Thank you for the good point!

The 24 is the correct reference number, and the number was changed by adding a reference.

  1. To assess the MVIC they do 3 repetitions, but they do not indicate the rest time between them.

-> Thank you for the good point. Authors made the following modifications.

MVIC data were collected for 5 s, of which 3 s was used for normalization (excluding the first second and the last second); a 5-second measurement and a 5-second break were repeated three times, and the maximum value was used for analysis [29]. (The 181-4th row

  1. References to Figure 6 are superfluous (lines 207, 210 and 213).

->Thank you for your good point.
deleted reference according to your opinion.

  1. There is a lack of a reference to guide the treatment of MLD applied (lines 213-214)

-> Thank you for the good point. Authors added the following modifications.

Reference: Wittlinger H, Wittlinger D, Wittlinger A, and Wittlinger Maria. Dr. Vodder`s Manual Lymph Drainage: A Practical Guide. Thieme.pp.62-65.

  1. Table 2 seems to have been made only with the data that was significant, is this the case? Why?

->There are lots of data in kinematic data during mSEBT, so authors only described data for significance

14.The References section must be adapted to the journal's standards: journal name in abbreviated italics and year of publication in bold.

-> Thank you for your good point. referance changed according to journal's standard.

  1. Only 9/42 are references from the last 5 years. Could it be more up-to-date?

-> Due to the majority of studies on Manual Lymphatic Drainage (MLD) focusing on cancer patients, there is a scarcity of recent research directly relevant to this study..

Reviewer 3 Report

Comments and Suggestions for Authors

The authors provide an analysis of the impacts of manual lymphatic drainage, mobilization with movement, and myofascial release on dorsiflexion range of motion and balance ability.  While these results represent an important contribution to the field, there are several clarifications that are required to address this topic more strongly:

1)     The introduction lacks strong transitions among the various techniques listed throughout.  Additionally, it is mentioned that the authors expect the application of MLD to “promote drainage of wastes in the body that may occur when MWM and MFR are performed…” however, no measures of waste drainage were incorporated into this study.  This is not appropriate to include in your hypothesis, but rather should be included in your discussion as a possible explanation of results.

2)     Methods – it would be beneficial to provide more substantial explanations of the test procedures/techniques used in this study.

3)     Methods – what were the qualifications/training of the researcher administering the test procedures?

4)     Was there any familiarization with the modified lunge test prior to collecting the data?  Were there any parameters put in place for an ‘acceptable’ effort during the test?  For instance, what if somebody’s third effort was significantly different than efforts 1 and 2?

5)     In the first paragraph of the Discussion, it is stated that these results established your hypothesis.  This may be an issue of using an incorrect word, but your results should not be establishing a hypothesis (Line 258).  The data should be supporting or refuting an already-made hypothesis.  Please clarify your intent with this statement.

6)     Limitation of this study includes lack of blinding of the assessors for your outcome – a stronger design would have been to blind the assessors and have independent research assistants administer the actual test procedures/techniques to each participant.

7)     How might the age of your participants limit the generalizability of these data as well?  One might assume that the orthopedic limitations/muscle shortening that may benefit from these techniques may inherently worsen with age, so a potential cohort (older individuals) that may directly benefit from these protocols have not been discussed.

8)     In your conclusions, you mention that “the application of MLD after MWM and MFR effectively discharged substances in the interstitium that may have accumulated after these procedures.”  However, no actual measures of the clearance of these substances was performed, and no blinding of assessors was incorporated.  Therefore, this conclusion is too strong for these data.  Please revise to include any statements of potential mechanisms of  improvement into the Discussion, with an acknowledgement that future studies are required to further explore this potential mechanism.

9)     The conclusion includes re-wording of what was already stated in the Discussion – it would be more effective to titrate the conclusion down to the few key points and how clinicians may use these findings for the direct benefit of patients.

Comments on the Quality of English Language

n/a

Author Response

Thank you for the reviewer. 

It helped a lot to make better this paper. 

Authors attached the file to response. 

The authors provide an analysis of the impacts of manual lymphatic drainage, mobilization with movement, and myofascial release on dorsiflexion range of motion and balance ability. While these results represent an important contribution to the field, there are several clarifications that are required to address this topic more strongly:

1)The introduction lacks strong transitions among the various techniques listed throughout.Additionally, it is mentioned that the authors expect the application of MLD to “promote drainage of wastes in the body that may occur when MWM and MFR are performed…” however, no measures of waste drainage were incorporated into this study.This is not appropriate to include in your hypothesis, but rather should be included in your discussion as a possible explanation of results.

       Authors deleted the hypothesis and the reference

2)Methods – it would be beneficial to provide more substantial explanations of the test procedures/techniques used in this study.

-> thank you for the good point. Authors added the Test procedures(line202- line220) .

3)Methods – what were the qualifications/training of the researcher administering the test procedures?

-> Added the qualification.

MLD was applied by a researcher who has undergone over 40 hours of training, and to minimize variations in technique, only one applied MLD throughout this study.

4) Was there any familiarization with the modified lunge test prior to collecting the data?Were there any parameters put in place for an ‘acceptable’ effort during the test?For instance, what if somebody’s third effort was significantly different than efforts 1 and 2?

->Thank you for the good point. Authors added the following modification.

Participants underwent two trials of the modified lunge test to familize themselves before DFROM measurements were taken. .

5)In the first paragraph of the Discussion, it is stated that these results established your hypothesis.This may be an issue of using an incorrect word, but your results should not be establishing a hypothesis (Line 258).The data should be supporting or refuting an already-made hypothesis.Please clarify your intent with this statement.

Thank you for the good point. Authors Authors changed the sentence.

This study aimed to find out which intervention application had the most positive effect on the study results in terms of DFROM and dynamic balance ability when comparing the MR, MD, and MRD groups. As a result of the study, the hypothesis of this study was supported that the application of MWM and MLD were performed, and that the MRD group would be the greatest improvement.

6)Limitation of this study includes lack of blinding of the assessors for your outcome – a stronger design would have been to blind the assessors and have independent research assistants administer the actual test procedures/techniques to each participant.

->Thank you for the good point.

          Authors added the limitation, which we will consider in further study.

7)How might the age of your participants limit the generalizability of these data as well?One might assume that the orthopedic limitations/muscle shortening that may benefit from these techniques may inherently worsen with age, so a potential cohort (older individuals) that may directly benefit from these protocols have not been discussed.

-> Thank you for the good point. Added the limitation.

8)In your conclusions, you mention that “the application of MLD after MWM and MFR effectively discharged substances in the interstitium that may have accumulated after these procedures.”However, no actual measures of the clearance of these substances was performed, and no blinding of assessors was incorporated.Therefore, this conclusion is too strong for these data.Please revise to include any statements of potential mechanisms ofimprovement into the Discussion, with an acknowledgement that future studies are required to further explore this potential mechanism.

->Thank you for the good point.

Authors  Changed the conclusions.

9)The conclusion includes re-wording of what was already stated in the Discussion – it would be more effective to titrate the conclusion down to the few key points and how clinicians may use these findings for the direct benefit of patients

-> Thank you for the good point.

Authors  Changed the conclusions as following:

This study assessed changes in DFROM in the modified lunge position, as well as  kinematic and kinetic data during mSEBT when MR, MD, and MRD groups were randomly assigned to healthy adults with short calf muscles, and interventions were applied. When compared to the MD group, the MRD group showed a significant increase in DFROM and a significant decrease in pelvic rotation in the PostM forward direction, while compared to the MR group, the MRD group showed a significant increase in mGCM activity in the PostM backward direction. Therefore, it is suggested that the MLD with MWM and MFR technique could be applied to increase DFROM and improve dymatic balance in the PostM direction in young healthy adults with short calf muslces.

Reviewer 4 Report

Comments and Suggestions for Authors

Review on the article Effects of manual lymphatic drainage with mobilization and myofascial release on kinematic, kinetic data and muscle activities during dynamic balance in adults with calf shortening.

Thank you for the opportunity to review the manuscript. I have listed comments & suggestions and hope they will improve the overall quality of the manuscript.

Title: I think that calf shortening might be understood as body part shortening, so I'd suggest inserting muscle to become " calf muscles shortening".

Introduction: There is citation: "Manual lymphatic drainage (MLD) is a treatment technique that reduces inflammation, edema, and pain by relaxing sympathetic nerves without irritation through mild skin elongation and by promoting drainage of body fluids and waste in the interstitium and lymph vessels" which I think is not related to the topic as study participants are relatively  healthy young people only with shortened muscles and it is not related to inflammation because of breast cancer. These two [10 and 12] references should be removed from the list. 

The hypothesis is unclear: The hypothesis of this study is believed that the application of MLD will promote drainage of wastes in the body that may occur when MWM and MFR are performed, showing the greatest increase in the ankle joint DFROM and dynamic balance ability" how these wastes are related to increased ROM. 

Methods: line 92 - I do not understand this sentence. Could you please clear it up. 

Table 1 needs comparisons between groups at baseline in terms of all characteristics. 

Figure two is very big compared to other figures. All figures should be the same proportions. 

Line 206. Please correct this sentence as it is expression "your feet". 

Line 209. "The myofascial was relaxed". Myofascial needs a noun. 

Lines 210 and 213. Figure 6 is mentioned, but there is no Figure 6 in the article. 

Conclusions are not formulated based on results. It is only prediction that "The application of MLD after MWM and MFR effectively discharged substances in the interstitium that may have accumulated after these procedures". 

In general sample size was very small. This study might be considered as a pilot study. 

Reference list: There is another one reference [15] that is not related to study participants. As I understand study participants were not involved in any sporting activity. 

Reference list does not meet the requirements of this journal. 

Was the intervention procedure a one-off? Is it an instant inquiry. If not, what was the duration, frequency, intensity of the different interventions.

Author Response

Thank you for the reviewer. 

It helped a lot to make better this paper. 

Authors attached the file to response. 

Review on the article Effects of manual lymphatic drainage with mobilization and myofascial release on kinematic, kinetic data and muscle activities during dynamic balance in adults with calf shortening.

Thank you for the opportunity to review the manuscript. I have listed comments & suggestions and hope they will improve the overall quality of the manuscript.

1.Title: I think that calf shortening might be understood as body part shortening, so I'd suggest inserting muscle to become " calf muscles shortening".

-> Thank you for the good point. Added the word.

Effects of manual lymphatic drainage with mobilization and myofascial release on dorsiflexion range of motion, kinematic data and muscle activities during dynamic balance in adults with calf muscle shortening

2.Introduction: There is citation: "Manual lymphatic drainage (MLD) is a treatment technique that reduces inflammation, edema, and pain by relaxing sympathetic nerves without irritation through mild skin elongation and by promoting drainage of body fluids and waste in the interstitium and lymph vessels" which I think is not related to the topic as study participants are relatively  healthy young people only with shortened muscles and it is not related to inflammation because of breast cancer. These two [10 and 12] references should be removed from the list. 

-> Authors deleted the reference and   added another hypothesis

3.The hypothesis is unclear: The hypothesis of this study is believed that the application of MLD will promote drainage of wastes in the body that may occur when MWM and MFR are performed, showing the greatest increase in the ankle joint DFROM and dynamic balance ability" how these wastes are related to increased ROM.

 -> The hypothesis of this study was that the application of MLD and MFR would increase the softening of connective tissue in the body, potentially occurring when MWM and MFR were performed, thereby showing the greatest increase in ankle DFROM and dynamic balance ability

  1. Methods: line 92 - I do not understand this sentence. Could you please clear it up. 

-> Thank you for the good point. Authors changed the sentence.

Inclusion criteria included patients who had a passive ankle dorsiflexion with range of less than 8° while the knee was extended in the prone position; who, after adjusting for the neutral position under the subtalar joint, showed an ankle dorsiflexion with the knee flexion greater than that with knee extension by 5° or more; and in whom the difference in the length of both legs was less than 2cm

  1. Table 1 needs comparisons between groups at baseline in terms of all characteristics.

-> Thank you for your good point. Authors added p-value.

The analysis of variance (ANOVA) was performed to assess the general characteristics across the three groups, and it indicated that there were no statistically significant differences.

6.Figure two is very big compared to other figures. All figures should be the same proportions. 

-> Thank you for your good point. The picture size was corrected

7.Line 206. Please correct this sentence as it is expression " your feet".

-> Thank you for the good point. Corrected the sentence

8.Line 209. "The myofascial was relaxed". Myofascial needs a noun. 

-> Thank you for the good point. Corrected the sentence.

In MFR, it was relaxed by placing a tennis ball on the plantar fascia and calf muscle fascia and then applying pressure with weight

9.Lines 210 and 213. Figure 6 is mentioned, but there is no Figure 6 in the article.

-> Figure 6 was added along with an explanation of the intervention for each group.

10.Conclusions are not formulated based on results. It is only prediction that "The application of MLD after MWM and MFR effectively discharged substances in the interstitium that may have accumulated after these procedures". 

->Figure 6 was added along with an explanation of the intervention for each group.

11.In general sample size was very small. This study might be considered as a pilot study. 

-> authors also agree with your opinion. A small sample size is a limitation of this study, and future research will consider this point and pay attention to the subjects

12.Reference list: There is another one reference [15] that is not related to study participants. As I understand study participants were not involved in any sporting activity. 

-> Authors deleted the reference.

  1. Reference list does not meet the requirements of this journal.

-> It has been revised to adhere to this journal

  1. Was the intervention procedure a one-off? Is it an instant inquiry. If not, what was the duration, frequency, intensity of the different interventions.

-> The intervention method in this study is one-time and evaluates immediate effects.

Added the sentence.

Round 2

Reviewer 2 Report

Comments and Suggestions for Authors

I believe that the authors have taken on board many of the suggestions and recommendations made by the reviewers and have improved the manuscript. The work is interesting, but it has limitations that the authors themselves recognise and that limit the significance of the findings obtained.

Considerations to this redrafting of the article:

- the authors say that the study on which the sample size calculation is based is a pilot study that we do not know about, an effect size of medium magnitude is selected and the calculated minimum sample size is not reached.

- I did not see the new version of Figure 1.

- I would change the wording of the Conclusions section. This seems to me to be more appropriate for the Results section.

Comments on the Quality of English Language

I consider the English Language used to be correct, but I am not an expert in this language to judge it properly.

Author Response

Dear. reviewer

Thank you for your valuable feedback. 

The authors have made the following modifications to the paper: 

1) Line 397-398. Additionally, this study was a pilot study in which the calculated minimum sample size was not reached. 

2) The authors have made a new version of figure 1. 

3) I agree with you. The authors have changed the conclusions.

Line 402-412)

This study assessed changes in DFROM in the modified lunge position, as well as kinematic and kinetic data during mSEBT when MR, MD, and MRD groups were randomly assigned to healthy adults with short calf muscles, and interventions were applied.

The MRD group demonstrated significant and positive changes in muscle activity and maximal reaching distance during the mSEBT. Consequently, the application of MWM, MFR and MLD could be effective for improving PL and mGCM activity to enhance dynamic balance during the mSEBT among young healthy adults with shortened calf muscles. Since the sample size was small in this study, it is thought that more reliable results will be obtained if the same experiment is conducted with an increased sample size in the future.

Reviewer 3 Report

Comments and Suggestions for Authors

1)     On page 4 of 16 – “familiarize” is spelled incorrectly (line 146).

2)     The Conclusion contains a lot of results and is worded like an abstract – this would be greatly strengthened by focusing specifically on what key messages should be taken from this manuscript.  Current lines 413-415 are a great starting point to highlight the clinical significance of these data.  Lines 405-4-13 do not fit well with the Conclusion portion of this paper.

Author Response

Dear Reviewer,

Thank you for your valuable feedback.

The authors have made the following modifications to the paper: 

1) The authors have changed the words. familize --> familiarize (Line 145) 

2) The authors agree with you to modify the conclusions. 

Line 402-411) 

This study assessed changes in DFROM in the modified lunge position, as well as kinematic and kinetic data during mSEBT when MR, MD, and MRD groups were randomly assigned to healthy adults with short calf muscles, and interventions were applied.

The MRD group demonstrated significant and positive changes in muscle activity and maximal reaching distance during the mSEBT. Consequently, the application of MWM, MFR and MLD could be effective for improving PL and mGCM activity to enhance dynamic balance during the mSEBT among young healthy adults with shortened calf muscles. Since the sample size was small in this study, it is thought that more reliable results will be obtained if the same experiment is conducted with an increased sample size in the future.